# The rapid evolution of lungfish durophagy

Xindong Cui [1,2,3], Matt Friedman [4], Tuo Qiao[1,2], Yilun Yu[1,2,3] & Min Zhu [1,2,3✉]

Innovations relating to the consumption of hard prey are implicated in ecological shifts in marine ecosystems as early as the mid-Paleozoic. Lungfishes represent the first and longest-ranging lineage of durophagous vertebrates, but how and when the various feeding specializations of this group arose remain unclear. Two exceptionally preserved fossils of the Early Devonian lobe-finned fish *Youngolepis* reveal the origin of the specialized lungfish feeding mechanism. *Youngolepis* has a radically restructured palate, reorienting jaw muscles for optimal force transition, coupled with radiating entopterygoid tooth rows like those of lungfish toothplates. This triturating surface occurs in conjunction with marginal dentition and blunt coronoid fangs, suggesting a role in crushing rather than piercing prey. Bayesian tip-dating analyses incorporating these morphological data indicate that the complete suite of lungfish feeding specializations may have arisen in as little as 7 million years, representing one of the most striking episodes of innovation during the initial evolutionary radiations of bony fishes.

[1] CAS Key Laboratory of Vertebrate Evolution and Human Origins, Institute of Vertebrate Paleontology and Paleoanthropology, Chinese Academy of Sciences, 100044 Beijing, China. [2] CAS Center for Excellence in Life and Paleoenvironment, 100044 Beijing, China. [3] University of Chinese Academy of Sciences, 100049 Beijing, China. [4] Museum of Paleontology and Department of Earth and Environmental Sciences, University of Michigan, Ann Arbor, MI 48109, USA. ✉email: zhumin@ivpp.ac.cn

**M**ajor changes in mid-Paleozoic aquatic ecosystems have been attributed to the expanded dietary breadth of jawed vertebrates[1,2] following their first definitive appearance in the Silurian[3]. Paleobiological hypotheses implicate innovations in durophagy—the consumption of prey items protected by hard shells or exoskeletons—as drivers of ecological and evolutionary shifts in marine settings over the Phanerozoic[4–6]. Durophagy evolved many times in parallel among jawed vertebrates, including several groups of fishes[7–10] and tetrapods[11–13]. Apart from ambiguous evidence from the Silurian[14], the Early Devonian (late Lochkovian, ~415 Ma) sarcopterygian *Diabolepis* represents the first unambiguous vertebrate durophage[15], predating the oldest examples in "placoderms"[9], chondrichthyans[7], actinopterygians[16], and tetrapods[12] by millions of years. This early lungfish shows functional hallmarks of the dipnoan body plan, including toothplates formed from non-shedding dentition arranged in radial rows, a palatal bite, short jaws, and a firmly united mandibular symphysis[15,17,18]. Such features set *Diabolepis* and other lungfishes apart from plesiomorphic dipnomorphs including *Powichthys* and porolepiforms[15,19,20]. However, the timing and sequence of the origins of these and other key feeding innovations remain unclear due to the incomplete fossils of the closest relatives of lungfishes.

*Youngolepis praecursor* is a common member of the diverse sarcopterygian assemblage from the Lower Devonian Xitun Formation of Yunnan, China[21,22] and the Bac Bun Formation of Trang Xa, Vietnam[23]. As the first lobe-fin described from this fauna[21–25], *Youngolepis* and its peculiar combination of characters fueled a major renaissance in sarcopterygian systematics[26–33]. Current consensus places *Youngolepis* (sometimes together with *Powichthys*) within Dipnomorpha[31] at the base of the lungfish lineage, branching between the Porolepiformes and *Diabolepis*[34]. *Youngolepis* is therefore a pivotal taxon for investigating the origin of lungfish anatomical specializations. Many of the best-known anatomical regions of *Youngolepis*, such as the mandible and other external dermal bones, agree broadly with those of other early non-dipnoan rhipidistians (the clade comprising Dipnomorpha and Tetrapodomorpha, the lungfish and tetrapod total groups)[23–27,35]. However, the skull shows architectural modifications that unite *Youngolepis* with lungfishes, including loss of the dermal and endoskeletal intracranial joints and the presence of neurocranial fossae interpreted as accommodating origins of jaw adductor musculature[27]. Although *Youngolepis* is not autostylic, many of its cranial features represent logical steps in the transition from hyostyly to autostyly, a character synapomorphic for Dipnoi. Taken together, these attributes hint that *Youngolepis* possessed a distinctive feeding mode anticipating the specialized arrangement of lungfishes. Nevertheless, the palate, hyoid arch, and branchial skeleton––all of which are important to the acquisition and processing of prey––remain unknown in *Youngolepis*.

Two well-preserved specimens (IVPP V28375 and V28376) of *Y. praecursor* from the same locality as the holotype show these key anatomical structures. Micro-computed tomography (μCT) reveals that the substantially reorganized palatal dentition, geometry, and suspension of *Youngolepis* are much more lungfish-like than previously anticipated, and differ conspicuously from primitive rhipidistian or osteichthyan conditions. The palatoquadrate has a short and deep postorbital region accompanied by a stocky, vertically oriented hyomandibula. The entopterygoid bears a thickened and expanded horizontal lamina with radially arranged teeth like the toothplates of *Diabolepis* and other more crownward lungfishes. These represent biomechanical transformations consistent with durophagy, a hallmark of the feeding mode of dipnoans. *Youngolepis* therefore illustrates critical early stages in the development of a trophic strategy retained by a major vertebrate lineage for over 415 million years.

## Results

### Description

*Palate*. V28375 preserves the left ectopterygoid and a pair of palatoquadrates, dermopalatines, and entopterygoids (Figs. 1 and 2a–f). The left palatoquadrate, entopterygoid, ectopterygoid, and dermopalatine of V28376 (Fig. 2f) are fully articulated. Together, the two specimens provide a complete picture of the palate of *Youngolepis*.

The palatoquadrate complex (comprising both dermal and endoskeletal components) has a short postorbital portion and long suborbital portion in lateral view, differing substantially from the proportions of other early bony fishes[36,37]. The postorbital region is anteroposteriorly narrow and dorsoventrally deep (Fig. 2a–f), and the adductor fossa (Fig. 2b, d) accounts for more than half of the overall length of the palate. Both the ascending process, which articulates with the braincase posterodorsal to the orbit, and the autopalatine process, which articulates with the ethmoid region, are well preserved (Fig. 2b–e). The region of the palatoquadrate near the expected articulation with the basipterygoid process is unmineralized, forming a conspicuous notch.

The posterodorsal margin portion of the palatoquadrate curves laterally, forming a thickened ridge that defines a broad surface for the origin of the adductor musculature in the adductor fossa (Fig. 2b–e). A well-developed, biconvex quadrate condyle at the posteroventral corner of the palate marks its articulation with the lower jaw. The inner side of the posterior blade of the palate bears a vertical trough representing the spiraculo-hyomandibular recess (Fig. 2c, e). In buccal view, sigmoid sutures and gaps (Fig. 2a, f) in

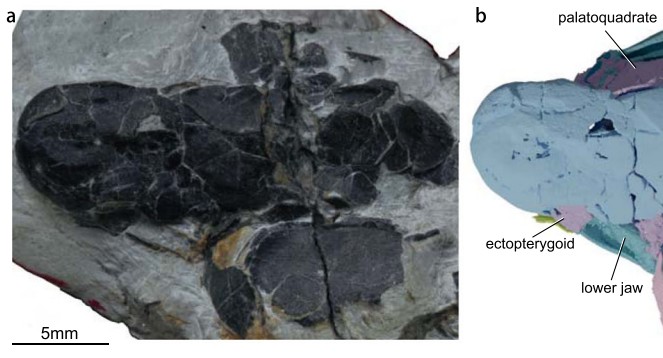

**Fig. 1 *Youngolepis praecursor*, specimen IVPP V28375 in dorsal view. a** Photograph and **b** virtual rendering.

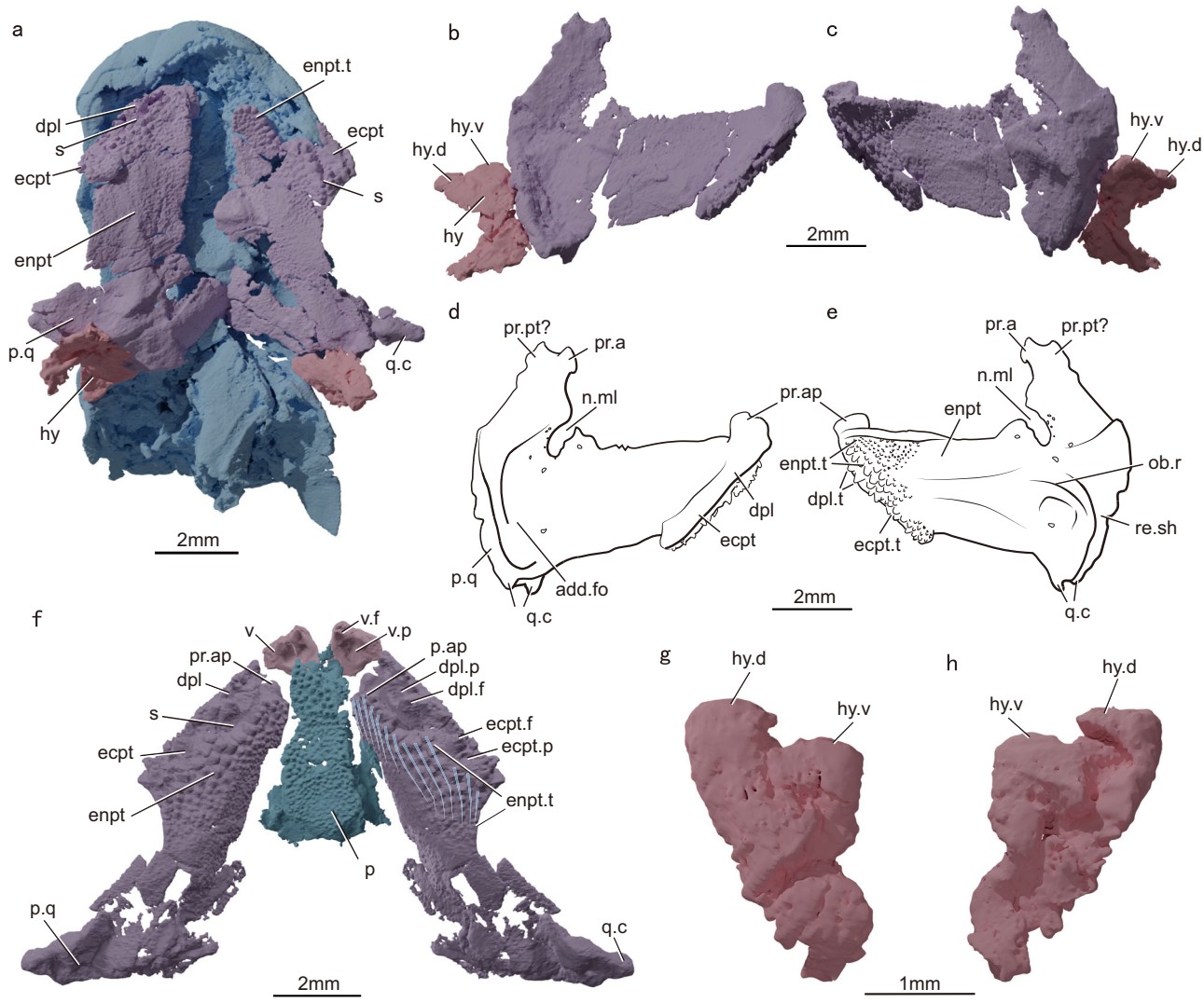

**Fig. 2 The palate and dorsal portion of the hyoid arch of *Youngolepis praecursor*. a** Virtual rendering of IVPP V28375 in ventral view showing palate and hyoid arch preserved in situ. **b**, **c** Virtual rendering of right palatoquadrate complex and hyomandibula of V28375 in lateral (**b**) and medial (**c**) view. **d**, **e** Interpretive drawing of right palatoquadrate complex of V28375 in lateral (**d**) and median (**e**) view. **f** Virtual rendering of palatoquadrate complex, vomer, and parasphenoid of V28376 in ventral view; palatoquadrate complex and vomer mirrored and restored to approximate life position. **g**, **h** Virtual rendering of right hyomandibula of V28376 in lateral (**g**) and median (**h**) view. add.fo adductor fossa, dpl dermopalatine, dpl.f dermopalatine fang, dpl.p dermopalatine fang replacement pit, dpl.t dermopalatine teeth, ecpt ectopterygoid, ecpt.f ectopterygoid fang, ecpt.p ectopterygoid fang replacement pit, ecpt.t ectopterygoid teeth, enpt entopterygoid, enpt.t entopterygoid teeth, hy hyomandibula, hy.d dorsal head of hyomandibula, hy.v ventral head of hyomandibula, n.ml notch for ramus of maxillaris and mandibularis, ob.r oblique ridge, p parasphenoid, p.ap pars autopalatina, p.q pars quadrata, pr.a ascending process, pr.ap autopalatine process, pr.pt? paratemporal process, q.c quadrate condyle, re.sh spiraculo-hyomandibular recess, s suture, v vomer, v.f vomerine fang, v.p vomerine fang replacement pit.

patterns of dentition mark the boundaries between the three dermal bones that line the inner surface of the palate: entopterygoid, ectopterygoid, and dermopalatine. The dermopalatine bears a single stocky fang and associated replacement pit (Fig. 2f; Supplementary Fig. 1a) flanked laterally by numerous tiny teeth (Fig. 2e, f) along the outer edge of the bone. The ectopterygoid (Fig. 2a–f) is shorter than the dermopalatine, but also bears a stout fang (Fig. 2f). As the largest dermal bone of the palate, the entopterygoid comprises two laminae with different orientations and contrasting patterns of dentition (Fig. 2c, e, f). The more medial lamina (Fig. 2c, e, f) is thin in cross-section, vertically oriented, and bears a shagreen of small denticles similar to those of most early bony fishes. By contrast, the expanded horizontal lamina (Fig. 2f) is greatly thickened in cross-section, and bears well-developed, conical teeth that are deployed in a series of precisely patterned, radiating rows. Thirteen rows are present, each of which contains between 5 and 10 teeth (Fig. 2f). Within each row, teeth increase in size toward the lateral edge of the entopterygoid (Fig. 2f).

*Hyoid arch.* The hyoid arch consists of the hyomandibula, ceratohyal, and hypohyal (Figs. 2a–c, g, h; 3). Both hyomandibulae are preserved in V28375 (Fig. 2a–c) and the right one is preserved in V28376 (Fig. 2g, h); all share the same inverted triangular shape. The distal parts of hyomandibulae in V28375 are only shown vaguely in scans, possibly due to incomplete mineralization, and cannot be segmented. Previous studies[24,27] permit restoration of the articulation between the hyomandibula and braincase. The hyomandibula has two swollen proximal heads (Fig. 2g, h), one dorsal and one ventral, corresponding to matching articular areas on the otic region of the braincase. The dorsal head is stubby and located on the posterolateral side,

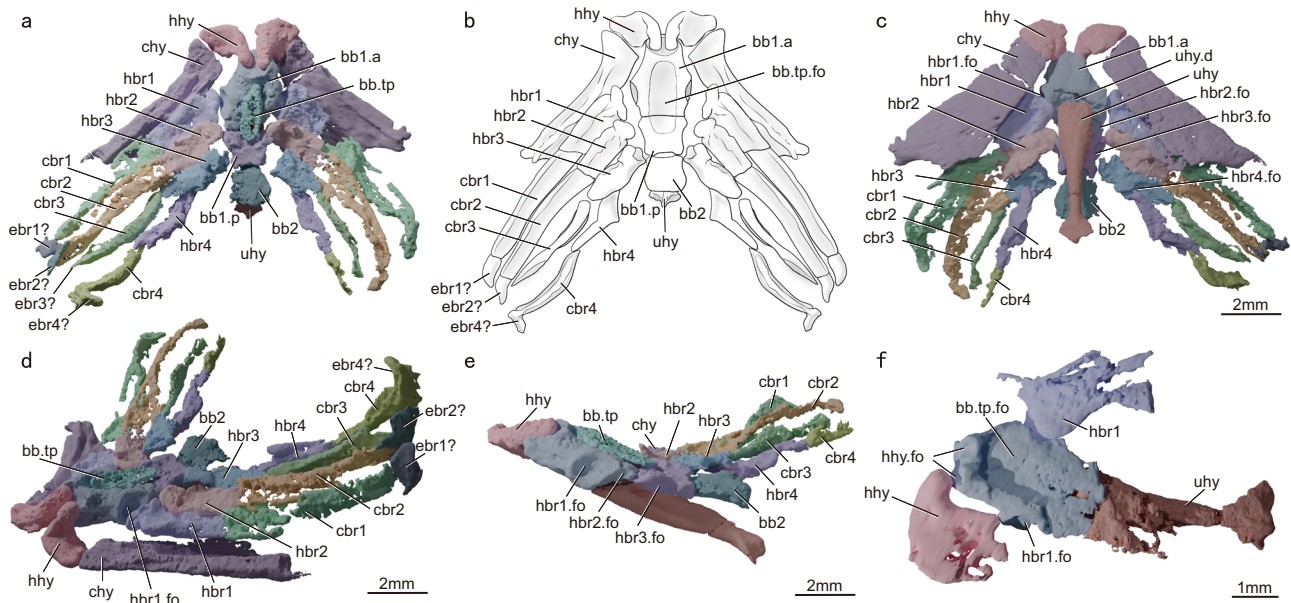

**Fig. 3 Ventral portion of the hyoid arch and gill skeleton of *Youngolepis praecursor*. a** Virtual rendering of V28375 in dorsal view. **b** Interpretive drawing of V28375 in dorsal view. **c** Virtual rendering of V28375 in ventral view. **d** Virtual rendering of V28375 in antero-dorsal-lateral view. **e** Virtual rendering of V28375 in lateral view. **f** Virtual rendering of V28376 in dorsal view. bb1.a anterior half of basibranchial 1, bb1.p posterior half of basibranchial 1, bb2 basibranchial 2, bb.tp basibranchial toothplate, bb.tp.fo fossa for the basibranchial toothplate, cbr1–4 ceratobranchials 1–4, chy ceratohyal, ebr1–4 epibranchials 1–4, hbr1–4 hypobranchials 1–4, hbr1–4.fo fossae accommodating hypobranchials 1–4, hhy hypohyal, hhy.fo two fossae accommodating the hypohyals, uhy urohyal, uhy.d depression for articulation with urohyal.

whereas the ventral head is slender and located on the anteromedial side (Fig. 2g, h). The hyomandibula tapers distally in V28376. Its inner face (Fig. 2h) is somewhat concave whereas the outer face (Fig. 2g) is slightly convex. The posterior margin of the hyomandibula bears a notch (Fig. 2g, h) possibly marking the path of the hyomandibular branch of the facial nerve[38]. The presence of accessory bones (e.g., interhyal) linking the dorsal and ventral halves of the hyoid arch is unclear.

The ventral portion of the hyoid arch comprises the hypohyal and certatohyal. The hypohyal is strongly curved (Fig. 3). Its narrow end articulates with the anterior face of basibranchial 1; a much broader facet meets the anterior end of the ceratohyal (Fig. 3). The ceratohyal is flat and broad, and bears a thickened ridge along its dorsolateral margin.

*Gill skeleton.* Specimen V28375 (Fig. 3a–e) preserves an articulated gill skeleton comprising the basibranchials, hypobranchials, ceratobranchials, and possible epibranchials. The basibranchial series includes three ossifications (Fig. 3a–c); the first two represent anterior and posterior halves of an incompletely mineralized basibranchial 1, while the last is basibranchial 2. Basibranchial 1 (Fig. 3a–c) forms an elongated hexagon, and its anterior margin bears an anterodorsally directed oval depression with two fossae (Fig. 3a, b, d–f) accommodating the hypohyals. A fossa (Fig. 3b, f) on the dorsal surface of basibranchial 1 accommodates an elliptical toothplate (Fig. 3a), and a shallow depression on the ventral surface marks the articulation with the urohyal (Fig. 3c, e). Basibranchial 1 bears three pairs of posterolaterally directed fossae (Fig. 3e). The first pair of fossae (Fig. 3e, f) are located immediately anterior to the unossified zone of basibranchial 1. These are deep, with well-defined edges, and accommodate the first pair of hypohyals (Fig. 3d–f). The second pair of fossae is located at the junction of the two ossified halves of basibranchial 1 and articulate with the second pair of hypobranchials (Fig. 3d, e). The third pair of fossae (Fig. 3d, e) are for the third pair of hypobranchials, and are located exclusively on the

posterior ossification of basibranchial 1. The concave posterior margin of basibranchial 1 articulates with a much smaller, trapezoidal basibranchial 2 (Fig. 3a, b, e) that bears no hypobranchials. The slender urohyal (Fig. 3c, e, f) extends beneath the basibranchials. It is flattened anteriorly, has a rod-shaped middle portion, and tapers posteriorly before terminating with a rhombic expansion (Fig. 3c, e, f) that lies in a horizontal plane.

*Youngolepis* possesses four pairs of hypobranchials and ceratobranchials (Fig. 3a–e). The first two hypobranchials (Fig. 3a–d) have prominent proximal heads that articulate with basibranchial 1. The proximal ends of the third pair of hypobranchials are poorly mineralized (Fig. 2a–d). The fourth pair of hypobranchials (Fig. 2a–d) are different from the first three pairs: they are smaller and have an "L" shape. The fourth hypobranchials (Fig. 2a–e) articulate with the third hypobranchials, rather than directly with the basibranchial series. The ceratobranchials are much longer than the hypobranchials (Fig. 3a–e). The first two pairs of ceratobranchials (Fig. 3a–d) are similar in shape, with tapered distal ends and wide proximal ends that articulate with the hypobranchials. Each bears a vascular groove (Fig. 3c) on its ventral surface. The third and fourth pairs of ceratobranchials (Fig. 3a–e) are smaller and more rod-shaped. Some small ossifications might represent epibranchials (Fig. 3a–e), but they are too poorly preserved to determine their identity or association with specific branchial arches.

**Inference of phylogenetic relationships and evolutionary timescale.** Consistent with previous analyses[27,31,33], our parsimony analysis and Bayesian tip-dating analyses place *Youngolepis* as the sister taxon of the clade comprising *Diabolepis* plus all more crownward lungfishes (Fig. 4a; Supplementary Figs. 2–17). Support for this arrangement is high, with a Bremer support value of 4 and a Bayesian posterior probability of 1. All sampled porolepiforms are resolved as a clade in all analyses, and this group plus *Powichthys* represent the sister lineage of all other dipnomorphs. Tip-dating

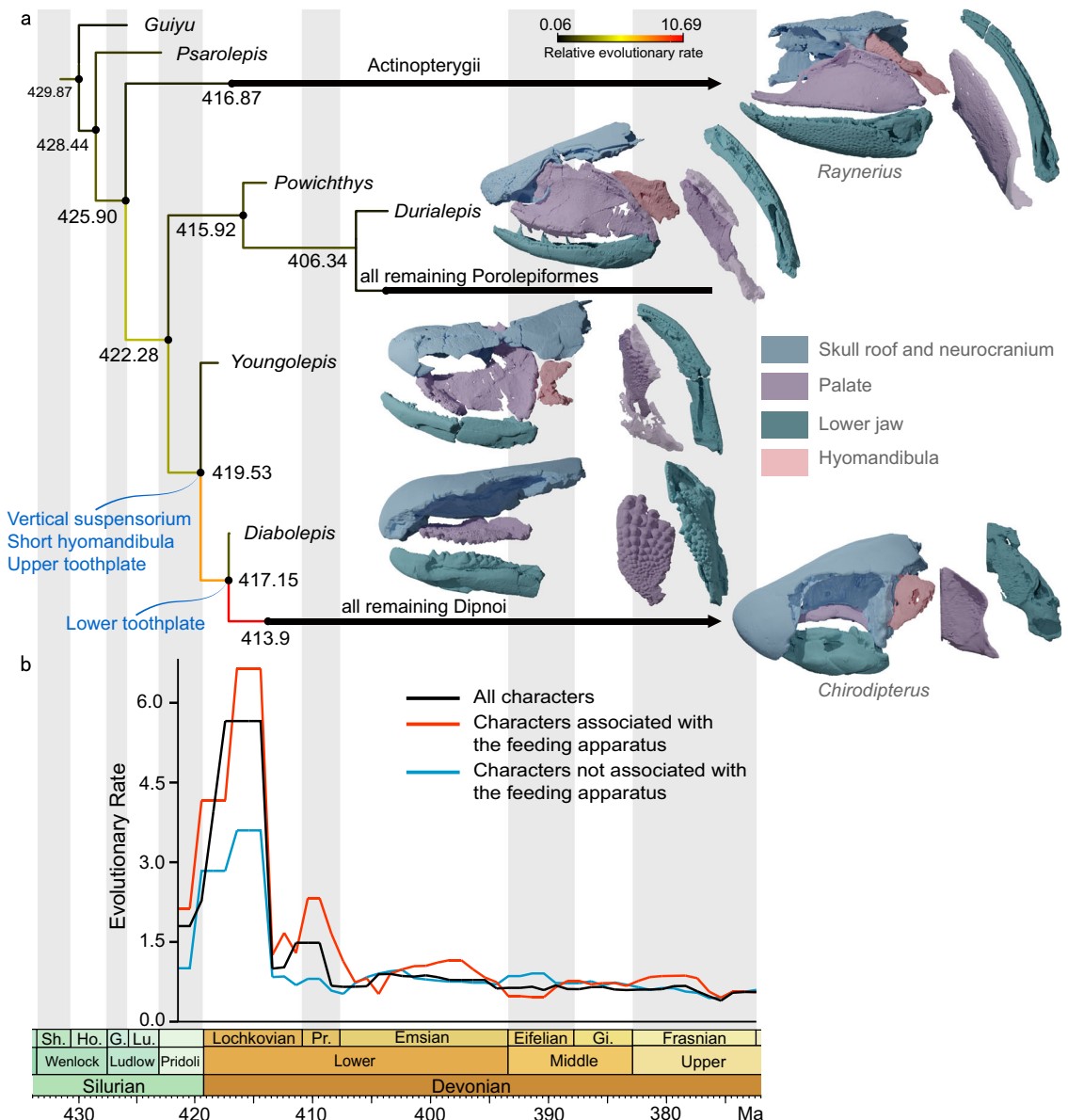

**Fig. 4 Timing of divergences and rates of trait evolution in lungfishes and their close relatives. a** Time-calibrated phylogeny with collapsed terminals for clarity. Divergence times represent median ages from the posterior distribution of trees. Branch color represents mean clock rate on that branch, measured in relative changes per site per Ma (unpartitioned IGR model with topological constraint based on parsimony analysis). Renders of exemplar taxa show, from right to left, skulls in left lateral view, left palate in ventral view, and right mandible in dorsal view. *Raynerius* from data for ref. [49], *Durialepis* from data for ref. [93], and *Chirodipterus* from data for ref. [94]. *Youngolepis* (IVVP V28375 and V28376) and *Diabolepis* (V17918, V28420.1 and V28420.2) are new. Note that multiple sarcopterygian lineages are inferred to branch between actinopterygians and dipnomorphs. These are not included here or in our analysis, which instead focuses on relationships and patterns of character evolution within Dipnomorpha. **b** Rates of discrete character evolution for different anatomical regions (IGR model with topological constraint based on parsimony solution, reflecting partitioned and unpartitioned analyses). G. Gorstian, Gi. Givetian, Ho. Homerian, Lu. Ludfordian, Pr. Pragian, Sh. Sheinwoodian.

analyses based on independent gamma rate (IGR) model (for details see the "Methods" section) estimate the divergence of por- olepiforms and lungfishes at 422.28 Ma (95% credible interval: 426.34–418.88 Ma), *Youngolepis* and more crownward lungfishes at 419.53 Ma (95% credible interval: 423.53–416.99 Ma), and *Diabo- lepis* and more crownward lungfishes at 417.15 Ma (95% credible interval: 419.20–414.91 Ma) (Fig. 4a).

## Discussion
Many characteristic features of lungfishes relate to their feeding apparatus[39,40]. Our analysis of *Youngolepis* reveals the phylogen- etically earliest evidence for some of these modifications, gives clues

to their initial functional significance, and provides a timescale over which they likely evolved. This provides an enriched framework for understanding one of the most conspicuous aspects of ecomor- phological innovation during the early evolutionary radiation of sarcopterygians: the processing of hard prey items.

**Assembly of the lungfish feeding apparatus**. Many structural peculiarities of dipnoan skulls relate to durophagy, and build upon features suggestive of strong bites in dipnomorphs more generally[41]. These include reorientation of the palatoquadrate and its fusion with the neurocranium, a shortened and non- suspensory hyomandibula, and a palatal bite formed by upper

(entopterygoid) and lower (prearticular) dental plates. Characteristic changes of lungfishes are paralleled to varying degrees in other groups of durophagous fishes that arose later in the Paleozoic, including ptyctodont placoderms[9], holocephalan chondrichthyans[7], and eurynotiform actinopterygians[42]. Well-preserved Pragian-Emsian dipnoans such as *Uranolophus* and *Dipnorhynchus* show that these iconic aspects of lungfish anatomy were established before the end of the Early Devonian[43–46]. The Lochkovian *Diabolepis*[47,48], a contemporary of *Youngolepis*[24], already shows well-developed pterygoid and prearticular toothplates with radial rows of teeth and a broad palatal bite (Fig. 4a; Supplementary Fig. 1b). The dentition of *Diabolepis* matches those of more crownward lungfishes in both structure and development, and thus provides few clues as to how this specialized arrangement arose from a more generalized osteichthyan pattern. Neither the hyomandibula nor palatoquadrate is known in *Diabolepis*, although it is apparent that the latter was unfused to the neurocranium. The complete skulls of *Youngolepis* reported here provide evidence for the initial modifications of the feeding apparatus near the base of the lungfish tree, and suggest a functional model for the origin of dipnoan durophagy.

The palate of *Youngolepis* diverges conspicuously from more general osteichthyan conditions (Fig. 4a). In sarcopterygians[19,37,38] and actinopterygians[49], the palatoquadrate primitively has a small suborbital ramus and a long postorbital portion from which the adductors mandibulae originate. The palatoquadrate complex and dermal cheek bones define an opening for these muscles that constitutes less than half of the length of the palate. A long, rod-like hyomandibula braces the posterior portion of the palatoquadrate, and roughly traces the dorsal margin of the palate. The suspensorium is therefore obliquely oriented, with the jaw joint often lying near the level of the rear of the skull roof. *Youngolepis* shows a very different arrangement. The hind region of the palatoquadrate has a near-vertical orientation. It is less distinctly separated from the suborbital portion, giving the palate a more triangular shape with a long opening for the jaw adductor muscles. These geometric changes have important consequences: they indicate a more vertical orientation of jaw muscles and a larger cross-section of those adductors, and also permit muscle attachments directly on the neurocranium as well as on the palatoquadrate[27]. Collectively, these all point to a strong bite in *Youngolepis*. In association with a restructuring of the palate, the hyomandibula of *Youngolepis* is short and stout. This foreshadows the condition in lungfishes, and parallels other examples like tetrapods[11] and holocephalans[7] where the tight binding of the palate and the braincase relieves the hyomandibula of its suspensory role.

The palatal dentition of *Youngolepis* shows a striking combination of general traits with those resembling lungfish features (Fig. 4a). The entopterygoid of *Youngolepis* has an expanded horizontal lamina relative to that of outgroups, forming a broad, flat surface on the roof of the mouth. By contrast, porolepiforms and *Powichthys* bear numerous, small tooth plates that augment the entopterygoid in contributing to a flat buccal roof[41,50]. In *Youngolepis*, the entopterygoid bears well-defined radial tooth rows, with individual cusps increasing in size toward the anterolateral margin of the entopterygoid. This resembles the pattern of dentition on the entopterygoid toothplates of *Diabolepis*[15,17] and another unnamed lungfish from the Lochkovian of Vietnam[51]. Some enlarged, dorsally directed cusps on the prearticular form a narrow shelf opposing the entopterygoid plate in *Youngolepis*, but these do not show the conspicuous radial patterning apparent on the palate. By contrast, *Diabolepis* has broad, well-defined prearticular toothplates. The internal structure of the entopterygoid teeth is indistinct in our scans, but the geometric correspondence to lungfish toothplates implies a

similar developmental pattern in *Youngolepis*: the addition of non-shedding teeth at marginal loci as the bone grew[52]. However, in contrast to *Diabolepis* and more crownward lungfishes, the toothplates of *Youngolepis* occur in conjunction with well-developed entopterygoids and dermopalatines that bear fangs and associated replacement pits, indicating periodic shedding. Similar coronoid fangs complement those of the palate (Fig. 4a). These blunt, squat fangs differ from the sharply pointed fangs of generalized rhipidistians, and suggest a role in crushing and fracturing—rather than piercing and ripping—prey[53,54]. No such fangs are known in lungfishes, despite the presence of small marginal palatal bones sometimes interpreted as vomers, dermopalatines, or ectopterygoids[55].

In contrast to the palate and dorsal hyoid arch, the ventral portion of the hyoid arch and gill skeleton of *Youngolepis* do not differ substantially from those of lungfish outgroups. The gill skeleton closely resembles that of porolepiforms[36,56,57], differing most conspicuously in retaining multiple basibranchials as in lungfishes and tetrapodomorphs. Broad, plate-like ceratohyals suggest that suction—rather than ram—feeding was the principal mode of prey capture[58] in porolepiforms[36,56] and *Youngolepis*, matching what is known in living lungfishes[39,59].

Taken together, these features suggest distinctive food acquisition and processing in *Youngolepis*, providing a functional model for understanding the major trophic shift associated with the origin of lungfishes. Clues about ancestral feeding modes come from ecomorphological comparisons with living fishes[60], gut contents in articulated fossil sarcopterygians[61,62], and trace fossils of failed predation attempts[63]. The primitive rhipidistian feeding style likely consisted of prey capture by either suction or ram feeding facilitated by puncture with large palatal and mandibular fangs, followed by the consumption of prey whole. By contrast, extant lungfishes reduce both hard and soft prey using either grinding (*Neoceratodus*) or shearing (*Lepidosiren*, *Protopterus*) between upper and lower toothplates[39,40]. Devonian lungfishes display a remarkable range of dental morphologies suggesting a broad repertoire of feeding styles, but the reduction of hard food items represents a common theme[64]. The stout, blunt-tipped fangs of *Youngolepis* are consistent with crushing rather than piercing prey[53], and may have been the primary means of food processing with the small entopterygoid toothplate playing a more secondary role. This conjunction of oral structures more typically associated with contrasting feeding modes within a single organism is evocative of major dietary shifts in other vertebrate lineages (e.g., the purported co-occurrence of baleen and teeth in some early mysticete whales[65]).

**Rapid trophic divergence in the lungfish lineage.** The major lineages of crown sarcopterygians first appear in the Early Devonian fossil records, leading to inferences of rapid evolutionary change in the early history of several groups[66–68], including lungfishes[69]. Our quantitative analyses reinforce this hypothesis. The distinctive features of the lungfish body plan—including autostyly and upper and lower toothplates—likely evolved in a short ~7 Ma window between the latest Silurian and the late Lochkovian (Fig. 4b). Both *Youngolepis* and *Diabolepis* are subtended by very short branches (Fig. 4a), suggesting that the overall anatomy of these two taxa might be reasonable models of ancestral stages along the lineage leading to lungfishes. We find that rates of change for characters most intimately associated with feeding (i.e., dentition plus mandibular, branchial, and hyoid arches) are accelerated at the same time as other traits (Fig. 4b), but to an even greater degree. This agrees with qualitative assessments of lungfish evolution noting the early and substantial modification of cranial structure relative to more conservative

postcrania, which closely resemble those of other early rhipidistians[70]. Lungfishes showing conspicuous modifications to median fin and body geometry first appear in the Middle Devonian[71,72], with taxa approaching modern dipnoan anatomy in the latest Devonian[73]. This is consistent with theoretical[74] and empirical[75] studies suggesting that early stages of phenotypic divergence are concentrated in aspects of trophic morphology, including jaws and teeth.

Following their origin, lungfishes went on to inhabit a broad spectrum of aquatic habitats and became the most taxonomically diverse group of Devonian sarcopterygians[31]. This contrasts with the other principal lineage of dipnomorphs, Porolepiformes, which were characterized by morphological conservatism and remarkably low taxonomic diversity over their >50 Ma history[61]. It seems probable that the distinctive lungfish feeding apparatus—the first stages of which are apparent in *Youngolepis*—provided the group with major new ecological opportunities that promoted their celebrated early diversification[69,76].

## Methods

**Fossil specimens.** This research complies with all relevant ethical regulations. This study is based on two articulated specimens of *Youngolepis praecursor* (Fig. 1) (IVPP V28375 and V28376) housed at the Institute of Vertebrate Paleontology and Paleoanthropology (IVPP), Chinese Academy of Sciences, each of which consists of a nearly complete head and partial body. Both specimens are somewhat flattened, but individual bones are uncrushed. This material is collected by Min Zhu from the argillaceous limestone of the Xitun Formation (Early Devonian: late Lochkovian) in Qujing, Yunnan, China[22].

**Computed tomography.** Specimens were prepared mechanically using pneumatic air scribes and needles under microscopes at IVPP. Both were scanned using a Nikon XT H 225ST industrial μCT scanner at the CTEES facility, Department of Earth and Environmental Sciences, University of Michigan. Both specimens were scanned using a tungsten target and the following parameters: IVPP V28375, voltage = 111 kV, current = 110 μA, effective pixel size = 11.55 μm, exposure time = 4 s, projections = 3141, images per projection = 2; filter = 0.35 mm Cu; IVPP V23876, voltage = 110 kV, current = 110 μA, effective pixel size = 12.15 μm, exposure time = 2.83 s, projections = 3141, images per projection = 2, filter = 0.23 mm Cu. Tomographic data were segmented using the software Mimics 22.0 (Materalise, Leuven, Belgium), with images of models rendered in Blender[77].

**Phylogenetic analysis.** In order to assess the placement of *Youngolepis* and assess patterns of trait evolution in early dipnomorph phylogeny, we assembled a matrix based on morphological characters taken from four analyses: Challands et al. [78] for lungfishes, Schultze[79] for porolepiforms, Lu et al. [34] for sarcopterygians more generally, and Giles et al. [49] for actinopterygians. 10 new characters based on our observations were added, and the states of the characters range from 2 to 6. The matrix (Supplementary Data 1) includes a total of 88 taxa (2 stem osteichthyans, 3 actinopterygians, and 83 dipnomorphs). The character data entry and formation were performed in Mesquite 3.61[80]. All analyses were rooted on *Guiyu oneiros* following previous hypotheses that 'psarolepids' are stem osteichthyans[34,81]. A parsimony analysis was conducted in TNT 1.5[82]. All of the characters were equally weighted. The maximum number of trees was set to 20,000 in memory. We used a traditional search with 1000 replicates of Wagner trees using random additional sequences. The TBR branch swapping that held 20 trees per replicate was performed. Supports for the clade were evaluated by Bootstrap resampling (incorporating replacement) using standard absolute frequencies (1000 replicates) and Bremer decay indices retaining suboptimal trees up to 20 extra steps.

We also inferred the morphological tree under a maximum-likelihood approach using IQ-TREE[83] (Supplementary Software). We used the M_k_ model[84] as the substitution model (Jukes–Cantor-type model for morphological data, where M stands for "Markov" and _k_ for the number of possible states) and made sure to correct for acquisition bias (i.e., argument "MK + ASC" used). We also accounted for rate heterogeneity across characters by using the discrete Gamma model[85] with four rate categories (i.e., argument "G"). The best-scoring maximum-likelihood tree is provided in Supplementary Fig. 3. We used the SH-like approximate likelihood ratio test[86] and the ultrafast bootstrap with 1000 replicates to assess the support of the branching patterns estimated in the phylogeny. The results here are consistent with those of other analyses we present.

We performed Bayesian tip-dating analyses in Mrbayes 3.2.8[87] with the newly assembled morphological data matrix (Supplementary Software) in order to infer divergence times and rates of character evolution. Each analysis comprised two runs with four chains (one cold chain and three in heated chains). We first conducted analyses with an unpartitioned matrix. Analyses were performed both with and without topological constraints reflecting the strict consensus tree from

our parsimony analysis. We used the Markov variable model (Mkv)[84] with gamma-rate variation across all characters for the likelihood calculation. The independent gamma rate (IGR), independent lognormal (ILN) and autocorrelated-rates (TK02) models were used to allow for rate variation across branches (Supplementary Software). To derive the clock rate prior, we used an in-house R script (Supplementary Software) that utilized the best-scoring maximum-likelihood morphological tree as input, the estimated mean age for each taxon and an estimated root age, 430 Ma, based on first appearance age of the outgroup taxon[3]. We estimate the morphological distance from the tips to the root of the tree (i.e., the "path length") using the function "node.depth.edgelength", part of the "ape" R package[88,89]. Next, each path length was scaled by the difference between the estimated root age and the mean age for each tip. The first appearance and last appearance ages for each taxon were download from THE DEEP BONE[90] database, and the mean age for each taxon was estimated as the mean of these two values.

Finally, the mean value of the scaled path length was used as the mean of the clock rate, and the exponential distribution (exp (40)) with the same mean was set as the prior of the clock rate. We used the default setting (exp (1)) as the prior of rate variance parameters. The fossilized birth death model (FBD)[91] was used as the tree prior. We do not have meaningful prior information for parameters for net speciation, relative extinction and relative fossilization rates for lungfishes; we therefore used non-informative priors for each. We used a uniform distribution, U(0, 10), as the prior of net speciation rate and a beta distribution, Beta(1, 1) for the relative extinction and relative fossilization rates. Because the dataset did not include the extant taxa, we designated the geologically youngest taxa in our analysis, *Conchopoma gadiforme* and *Gnathorhiza serrata*, as "extant" for the purposes of defining sampling fractions, and sets the extant sampling probability to 1. The fossil ages obtained from THE DEEP BONE[90] database were assigned uniform priors. The root age had an offset exponential prior with a mean of 432.95 Ma and a minimum of 427.95 Ma, which were slightly older than the time range of the outgroup taxon (*Guiyu oneiros*)[3]. To examine any differences in patterns of evolutionary rate through time for different anatomical regions (Supplementary Software), we performed analyses using two data partitions: the first included mandibular arch, hyoid arch, gill skeleton, and teeth (112 characters: 82–116, 123–187, 266–277) and the second contained all remaining characters (165 characters: 1–81, 117–122, 188–265), based on model settings mentioned above. Each analysis was run 200 million iterations, and the first 30% samples were discarded as burn-in.

Convergence of parameters was identified using Tracer[92] (ESS > 200). The results are given in Supplementary Table 1. Unpartitioned and partitioned analyses using the AR model failed to converge. Analyses using other models did converge with the exception of the partitioned analysis using IGR with no topological constraints. Because all analyses under the AR model failed to converge, and all converged analyses show comparable patterns of relationships, evolutionary timescale, and rates of character evolution over time, we did not perform the stepping-stone sampling to estimate marginal model likelihoods. On this basis, we regard our inferences about those patterns as robust to variation in assumptions of clock models. We presented the results of IGR model in the main text figure (Fig. 4), and all converged results in the supplement (Supplementary Figs. 4–17).

**Reporting summary.** Further information on research design is available in the Nature Research Reporting Summary linked to this article.

## Data availability

The CT data and 3D models generated in this study have been deposited in the figshare database with the link: https://doi.org/10.6084/m9.figshare.15134253.v1. The phylogenetic data generated in this study are provided in the Supplementray Data and Software files. The additional notes generated in this study are provided in the Supplementary Information file. The life reconstruction of *Youngolepis praecursor* and the associated biota is provided as the Supplementary Figure 18.

## Code availability

The codes used for the study are provided in the Supplementary Software files.

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

## Acknowledgements

We thank M.M. Chang and M. Kolmann for discussions and suggestions; C.Y. Xiong for specimen preparation; Y.M. Hou and P.F. Yin for CT scanning; T. Challands and S. Henderson provided CT data for *Chirodipterus*; B. Choo for artistic life reconstruction; C. Zhang for the help in tip-dating analyses. This work was supported by the Strategic Priority Research Program of Chinese Academy of Sciences (XDA19050102 and XDB26000000, M.Z., X.C., F.M., T.Q., and Y.Y.), the National Natural Science Foundation of China (42130209, M.Z., X.C., T.Q., and Y.Y.), and the Joint PhD Training Program of University of Chinese Academy of Sciences (X.C.). This study includes data produced in the CTEES facility at University of Michigan, supported by the Department of Earth and Environmental Sciences and College of Literature, Science, and the Arts.

## Author contributions

M.Z. and M.F. designed the project. M.F and X.C. collected tomographic data. X.C. processed the tomographic data. X.C. and T.Q. assembled and analyzed the phylogenetic dataset. Y.Y. performed the tip-dating analyses. X.C. and M.F. interpreted the data and wrote the first draft of the manuscript. All authors discussed and commented on the final version of the manuscript.

## Competing interests

The authors declare no competing interests.
