## [Peer Review File · Nature Communications]

The rapid evolution of lungfish durophagyREVIEWER COMMENTS

Reviewer #1 (Remarks to the Author):

This paper describes the anatomy of two important specimens of a Devonian sarcopterygian, †Youngolepis praecursor using computed tomography to reconstruct the cranial skeleton. These specimens show remarkable details of the feeding apparatus that resemble some of the specializations of lungfishes (Dipnoi), a group that radiated extensively during the Devonian and that has persisted until today. The descriptions and anatomical figures are excellent, although I suggest some additional labeling to clarify where the resorption pits for the fang are located and to better indicate the radiating tooth rows on the palate.

I made many small comments and queries on the .pdf (copy attached). Perhaps my most important question concerns monophyly of †Porolepiformes, a term that is used throughout the text and in Figure 4. I am not aware of any paper definitively showing monophyly of the †Porolepiformes; I may be wrong, but I think it is an open question whether some "†Porolepiformes" might not be more closely related to Dipnoi than to other "†Porolepiformes." Until the monophyly of †Porolepiformes is demonstrated, I think the authors would be better off just stating that their comparisons were made to †Durialepis.

In a similar vein, I think the word Dipnomorpha needs to be included somewhere in the body of the paper because there seems to be little doubt about monophyly of dipnomorphs. A final small quibble about Figure 4 is that the rest of the Rhipidistia are not shown - its almost like the evolution of Osteichthyes ended with Dipnoi!! Not a problem for me, perhaps, because I think lungfishes might actually be the pinnacle of vertebrate evolution, but general readers will appreciate more context. You can insert a new branch for the "all remaining rhipidistians" between your branch for Actinopterygii and †Porolepiformes (if they prove to be monophyletic or †Durialepis if you opt not to comment on the monophyly of †porolepiforms in this paper).

William E. Bemis

Reviewer #2 (Remarks to the Author):

This manuscript provides the first extensive description of the internal cranial anatomy of the dipnomorph sarcopterygian Youngolepis, from the Lower Devonian of China and Vietnam, with particular reference to its relationships to lungfishes and the rise of durophagy in jawed vertebrates. As a whole, the manuscript is most interesting and very well executed, and will be welcomed by all researchers working on osteichthyan phylogeny. My comments below are limited to a few remarks:
L.17: Durophagy becomes widespread only Devonian placoderms, especially in Middle and Late Devonian pachyosteorhynchid arthrodiras, such as Mylostomatids. This is coeval with the rise of durophagous holocephalan chondrichthyans, which then will populate the Carboniferous marine ecosystems.

L.53: Youngolepis also occurs in the Lower Devonian of northern Vietnam

L. 79: expansive...isn't it rather 'expanded'?

L. 94.: I have one question regarding the branchial apparatus: is there any evidence for branchial "teeth" or denticles? Such minute teeth covering the gill arches are present in porolepiforms, but lacking in lungfishes, and not shown in the scans of Youngolepis illustrated here. Such a lack of branchial teeth might be one more synapomorphy of the two taxa, as branchial teeth are widely present in all other osteichthyans.

L. 219: Is it what Bjerring once called "hamuloquadrate"?

L. 223: Palatal bite is clearly present in coelacanth (including extant ones), and was probably already there in porolepiforms, as suggested by the numerous toothed platelets covering the palate in Glyptolepis and Powichthys (see e.g. Clément & Janvier 2004, fig.8)

L.268: OK, and the same formation of Vietnam yields toothplates that are almost identical to

Diabolepis

L.271: Interestingly, the prearticular of Glyptolepis shows an incipient radial patterning of the denticles, radiating from its posterior region (see Jarvik's monograph).

L. 695: it is rather 'medial' view.

Reviewer #3 (Remarks to the Author):

This manuscript provides a thorough explanation about the morphology of *_Youngolepis_* species and, if I have understood correctly, how its specific and unique features seem to help better understand how the feeding mode evolved in bony fishes. While the authors have extensively documented the morphology of this species, the methods used to infer the morphological phylogeny and the divergence times are vaguely explained. There is a substantial amount of information that should be present both in the main and the supplementary text in case anyone was to reproduce the analyses mentioned in this manuscript.

Below, you will find some suggestions with regards to the text as well as some questions I have about the methods used in this study. Hope that they are useful!

===

Line 15: Maybe write "Innovations *about*" instead of "Innovations *~of~*".

Line 18-19: Maybe write as "New and exceptionally preserved *_Youngolepis_* specimens, which lived during the Early Devonian, [...]". I think it reads better for a non-familiar audience if written that way.

Line 25: Did you mean "piecing the prey"?

Line 25: Rewrite to "total-evidence dating (TED) analysis". You might also want to add some citations here about previous TED analyses.

Line 26: When you say that you incorporate these new observations in a TED analysis, are the "observations" morphological data? If so, it might be good to clarify as either molecular and/or morphological data can be used to infer species divergence times in a TED analysis.

Line 49: "remain" in singular as it refers to "the timing and sequence".

Line 49: When you say "a lack of information", what do you mean? Lack of evidence in the fossil record, incomplete fossils from different taxa that lead to missing characters in morphological matrices, ...? Please clarify here.

Line 91: Just because I see you have been consistent with the oxford comma, you might want to add it here after "dermopalatines".

Line 196: Here, you are not only reporting the results you obtain when you infer a phylogeny (more specifically, a *morphological* phylogeny), but you are also reporting estimates of divergence times. Therefore, it would be fair to use a header such as "Morphological phylogeny inference and divergence-time estimation analysis". In addition, it would be good to mention the approach you have used to infer the morphological phylogeny and the divergence times so the reader can have an idea about what you have done before going to the "Methods" section.

Line 380: You say "characters", but character data can be either morphological or molecular. It is best if you write "morphological character data" or "morphological data" instead of just "characters" when you first introduce the concept to the reader.

Line 383: How many characters were removed and how many new characters were added based on your observations? You might want to add this information here. It would also be nice to include how many states the characters you include in the matrix have (e.g., maybe there are morphological characters with just 2 states and others with 5 states, so just write the range from the minimum to the maximum number of states one would expect to find in your morphological matrix).

Line 387: When using TNT, do you need to choose specific values for any parameters or do you need to choose specific options to run the software? If so, you might want to give more details about the setup that you have used to run this software to infer a morphological tree under parsimony so the user can reproduce your analysis.

In addition, I would like to see if using a maximum-likelihood approach to infer the morphological

phylogeny with your data (e.g., you can use either `RAxML-ng` or `IQ-TREE` for this analysis) converges with the estimated morphological tree inferred under parsimony. This is because the morphological matrix you have is not extremely large and an ML approach can run in a reasonable amount of time - besides, ML approaches have been widely used to infer both molecular and morphological phylogenies for the past years, more than parsimony has been used. For instance, if you choose to run `IQ-TREE`, you can follow the instructions [here](http://www.iqtree.org/doc/Tutorial) under the section "Binary, morphological and SNP data", where you will find the commands you need to run when using your morphological matrix as the input data. You are given very clear guidelines about how you can choose the model under which you will infer your morphological phylogeny as well as the commands you need to use to account for acquisition bias.

Line 391: Rewrite to "tip-dating analyses".

Line 392: "phylogenetic dataset" is not the correct term you should use here. You are using two input files for each analysis (according to the supplementary material that I see you have attached): a "morphological character matrix" and a "parsimony-inferred morphological phylogeny". These are the input data that MrBayes is using to infer the divergence times (apart from the calibrations I see you have included and the different priors you have set) under a Bayesian tip-dating approach. Also, note that the format of the files in which these data are stored is nexus.

Please, remove "phylogenetic dataset" and rewrite this bit accordingly.

Line 395: You write "The first did not apply any character partitioning, while the used four partitions". Do you mean that in "the first analysis" you did not partition your morphological character matrix while in "the second analysis" you partitioned it into four blocks based on anatomical systems? If this is what you meant, please rewrite this section so it is clearer for the reader. Also, it would be useful to know how many characters you included in each partition and the range of the states that each character had in each block (i.e., min-max range per block).

Specific comments about the "Methods" section:

- Why did you use `prset clockratepr = normal(0.01, 0.02)`? I mean, why did you choose that specific distribution for the prior on the rates? You should include in the methods section a justification for choosing a normal prior distribution with this mean and sd in the same way that you have included the details about the FBD prior you have used. The same applies for other priors you use when you run MrBayes.

- Why did you choose to run an ILN instead of an autocorrelated-rates model? It would have been nice to see a model selection analysis with which the best rate model (ILN or the AR) is selected according to the data (and then you can use that when you infer the divergence times).

- Are you using a specific fossil to set the exponential prior on the root age? If so, please specify which one so you can justify the age for the root.

- You have not confirmed if your MCMCs have converged. Common practice is to run each analysis at least twice (and long enough) to confirm that chains have converged. For instance, you can visually check if chains have converged by plotting the divergence times that you infer under the two runs and, if you see a line $x \approx y$, then the chains are likely to have converged. In addition, you should also verify that the effective sampling size (ESS) for each parameter inferred is $ESS \geq 200$. You can use R packages such as `coda` or `rstan` to verify this. You can also use the software `Tracer` for that purpose. If your chains have not converged, you might need to increase your sampling frequency while keeping the same number of samples to be collected (in that way, you do not increase the amount of disk space for the output file in which the collected samples are printed out but you are still running the chain longer). To calculate the sampling frequency and the length of the chain, you need to know that:

$length_chain = burnin + sampling_freq * num_samples$

- In your supplementary material, you have four different directories: `tip dating_constrained with strict consensus topology from parsimony analyses`, `tip dating_unconstrained`, `tip dating_4 partitions_constrained with strict consensus topology from parsimony analyses`, and `tip dating_constrained with majority consensus topology from parsimony analyses`. My questions and comments with regards to this zip file are the following:

a) Please rename the file names so there are no spaces (e.g., use underscores or dashes). It is then easier to deal with them if they are to be open from the terminal.

- b) Fix the typos in the file names, i.e., "tip dating".
- c) Even though you have figures with the output morphological phylogeny drawn in your supplementary material, you should include a file with the resulting timetree inferred by MrBayes in the supplementary material either in nexus or newick format (these are the common formats used to read the trees in software such as `FigTree` or to manipulate them in `R` or `Python`; among many other uses). It would also be nice to see the output files by MrBayes and know which seed number was used to run the chain so the analyses can be reproduced by the user.
- d) I see in your nexus input files for MrBayes that you have used specific calibrations for the tip-dating analysis. Why did you specifically use uniform densities as calibrations and not, say, softbound calibrations as most dating analyses do? What are the maximum and minimum ages of each calibration you include based on? You should have a justification for each calibration included in the analysis (e.g., age justification based on a specific taxon?). You can add this information in the supplementary material.
- c) In your ms, you talk about divergence-time inference when using a topological constraint, yet here you have two directories with input data that suggest you ran two analyses without constraints. Why don't you report the results you obtained with the "unconstrained" analyses here in the main text? I can see you have Supp. Fig. 5, but you do not mention anything in the main text or compare the estimates you get when the topology was constrained and when it was not. Are there any topological rearrangements? Are times older or younger if constraints are/are not applied? Etc.
- d) You provide an R script which should output some plots and CSV files, but I cannot see them in the supplementary information. Could you please attach them? Also, it would be nice if you provided the user with the input files you use to run this R script so they can reproduce your results.
- e) It would be interesting to plot the differences between the time estimates inferred when partitioning the data and when the data were not partitioned.

Responses to the three reviewers:

We thank the three reviewers for thoughtful comments, corrections, and suggestions. We have implemented these as far as possible, and believe that the modifications strengthen our contribution.

Reviewer #1

I suggest some additional labeling to clarify where the resorption pits for the fang are located and to better indicate the radiating tooth rows on the palate.

[Response]: Thank you for your suggestion. We added the labels of “vomerine fang” (v.f), “vomerine fang replacement pit” (v.p), “dermopalatine fang replacement pit” (dpl.p), and “ectopterygoid fang replacement pit” (ecpt.p). To better show these, we have also included enlarged versions of these bones in the Supplement Figure 1. In order to highlight the radiating tooth pattern of the entopterygoid, we have overlain guides on one side of the palate for Figure 2 in the main text.

Perhaps my most important question concerns monophyly of †Porolepiformes, a term that is used throughout the text and in Figure 4. I am not aware of any paper definitively showing monophyly of the †Porolepiformes; I may be wrong. but I think it is an open question whether some “†Porolepiformes” might not be more closely related to Dipnoi than to other “†Porolepiformes.” Until the monophyly of †Porolepiformes is demonstrated, I think the authors would be better off just stating that their comparisons were made to †Durialepis.

[Response]: It is correct that there has not been a phylogenetic analysis containing all the genera or species of Porelepiformes. Most analyses include the well-known genera *Porolepis* and *Glyptolepis* (Zhu et al., 2016; Lu et al., 2017; Brazeau et al., 2019). Our study builds upon these past efforts, adding putative porolepiform genera *Durialepis*, *Nasogaluakus*, *Holoptychius*, *Quebecius*, and *Laccognathus*. We recover these genera, plus *Porolepis* and *Glyptolepis*, as a clade in our various phylogenetic analyses (see figures in the supplement). We did not include some additional, poorly known porolepiforms like *Duffichthys* and *Heimenia*. We are reasonably certain that the core taxa considered by our study do, in fact, represent a clade. However, we note that our inferences about the implications of *Youngolepis* for understanding the origin of lungfish feeding structures are not sensitive to the monophyly (or non monophyly) of porolepiforms.

Brazeau, M. D. et al. Endochondral bone in an Early Devonian ‘placoderm’ from Mongolia. *Nature Ecology & Evolution*, 4(11), 1477-1484 (2020).

Lu, J. et al. A new stem sarcopterygian illuminates patterns of character evolution in early bony fishes. *Nature communications*, 8(1), 1-8 (2017).

Schultze, Hans-Peter. "Dipnoans as sarcopterygians." *Journal of Morphology* 190. S1: 39-74 (1986).

Zhu, M. et al. A Silurian maxillate placoderm illuminates jaw evolution. *Science*, 354(6310), 334-336 (2016).

I think the word Dipnomorpha needs to be included somewhere in the body of the paper because there seems to be little doubt about monophyly of dipnomorphs.

[Response]: We agree with the referee's suggestion. We have now introduced the term early in the manuscript, and use it several times after.

A final small quibble about Figure 4 is that the rest of the Rhipidistia are not shown - its almost like the evolution of Osteichthyes ended with Dipnoi!! Not a problem for me, perhaps, because I think lungfishes might actually be the pinnacle of vertebrate evolution, but general readers will appreciate more context. You can insert a new branch for the "all remaining rhipidistians" between your branch for Actinopterygii and †Porolepiformes (if they prove to be monophyletic or †Durialepis if you opt not to comment on the monophyly of †porolepiforms in this paper).

[Response]: We agree (both about the omission, and the status of lungfishes as the pinnacle of vertebrate evolution!). However, this would create some problems in our mind, most notably the "guesswork" of where in time the branch should go, plus the necessity to potentially include other sarcopterygian lineages (e.g., coelacanth, onychodonts) in similar fashion. Because we did not include these lineages in our analysis, we feel uncomfortable manually inserting them. As a solution, we have noted that the tree/analysis is abbreviated, and that multiple groups branch between actinopterygians and dipnomorphs.

L15: of for.

[Response]: Revised.

L29: radiation radiations.

[Response]: Revised.

L42: ~~placoderms~~ "placoderms".

[Response]: Revised.

L49: in about.

[Response]: Revised.

L58: Is †Porolepiformes a monophyletic group? Source 33 does not address this question, and I think there is reason to believe that some "†porolepiform" taxa are more closely related to lungfishes than to other "†porolepiforms." If the group is not monophyletic, or if its monophyly is questionable, then please put the name "†Porolepiformes" in quotes so as not to imply that we know they are monophyletic. If †Porolepiformes is monophyletic, then please include a source that confirms monophyly of †Porolepiformes. This is important because readers will want to know to which "†porolepiform" you are making comparisons, †Porolepis or †Holoptychius.

[Response]: See previous comment concerning the sampling of porolepiform taxa in this analysis.

L62: ~~lungfish~~-lungfishes.

[Response]: Revised.

L105: Perhaps call out where to best see this notch and give its abbreviation.

[Response]: This notch refers to the "notch for rr. maxillaris and mandibularis (n.ml)" in Figure 2.

L109: Add 'in the adductor fossae'.

[Response]: Added.

L109: Add 'quadrate'.

[Response]: Added.

L116: Is this labeled in the figure? I am not sure that I see it, so it would be best to label it or remove the call out to something not easily seen.

[Response]: The dermopalatine fang replacement pit (dpl.p) is now labeled in the main text figure, as well as in the higher resolution supplementary figure.

L119: Please label it - I am not sure that I am seeing it, and it is very important to the discussion of shedding versus non shedding dentitions.

[Response]: The ectopterygoid fang replacement pit (ecpt.p) is now labeled in the main text figure 2, as well as the higher resolution supplementary figure 1.

L120: delete '(Fig. 2c, e, f)'.

[Response]: Deleted.

L144: Do you know what passes through this notch? I have some ideas, but it would be good to state this if you can.

[Response]: The most plausible interpretation of this feature is that it accommodates the hyomandibular branch of the facial nerve (VII); we have added this to the text. Typically in bony fishes this branch is enclosed within a canal in the hyomandibular, but it is possible that the bone is incompletely mineralized in *Youngolepis*.

L148: Add 'portion of the'.

[Response]: Added.

L149: ~~The~~ Its narrow end of the ~~hypohyal~~ articulates with the anterior face of basibranchial 1, ~~while~~; a much broader facet meets the anterior end of the ceratohyal (Fig. 3).

[Response]: Revised.

L152: Maybe find a different word – I am having trouble interpreting what you mean by "trapeziform profile".

[Response]: We agree that this is confusing and adds little useful information. Since the bone is clearly shown in the figure, we decided to remove this potentially ambiguous descriptive text.

L169: are is.

[Response]: Revised.

L201: Which "†porolepiforms" - the group may not be monophyletic.

[Response]: See previous note about sampling of porolepiforms in this analysis.

L208: delete 'of the'.

[Response]: Deleted.

L213: ~~ecological~~ ecomorphological.

[Response]: Revised.

L217: Many structural peculiarities of the dipnoan ~~skull~~-skulls appear-related to durophagy.

[Response]: Revised.

L219: shortened.

[Response]: Revised.

L243: add 'bones'.

[Response]: Added.

L251: jaw adductors-muscles.

[Response]: Revised.

L262: This is a vague way of saying that they have some features that closely resemble those of lungfishes.

[Response]: We have used the more explicit term "resembling" here, as we agree the previous construction was vague.

L265: Please make sure that these are explicitly labeled somewhere.

[Response]: We added the labels of the radiating tooth rows on the entopterygoid in Figure 2f.

L278: Please be sure that these are explicitly indicated on the figures somewhere - I am sure that they are there, but I could not distinguish the resorption pits.

[Response]: We added the labels of "vomerine fang" (v.f), "vomerine fang replacement pit" (v.p), "dermopalatine fang replacement pit" (dpl.p), and "ectopterygoid fang replacement pit" (ecpt.p) in Figure 2.

L280: These blunt, squat fangs differ from the sharply pointed ~~examples in~~ fangs of generalized rhipidistians.

[Response]: Revised.

L286: Add 'portion of the'.

[Response]: Added.

L323: This is the first mention of autostyly. General readers are more likely to be familiar with this concept than with other features of lungfish anatomy, so maybe introduce the concept much higher in the paper with a sentence like, "Although †*Youngolepis* praecursor is not autostylic, many of its cranial features represent logical steps in the transition from hyostyly to autostyly, a character synapomorphic for Dipnoi."

[Response]: Thank you for your suggestion. We added this sentence in the Introduction.

L325: ~~*Diablepis*~~ *Diabolepis*.

[Response]: Revised.

L328: This is jargon – what exactly is a cranial partition? I can think of many things. Please be more precise about what you mean.

[Response]: We have simplified our comparative analyses, and so only consider rates of change in a subset of cranial features versus all remaining parts of the skeleton. This sentence no longer appears in our revised text.

L329: a in rates of change in the braincase and oral features (Fig. 4b). This is totally unclear to me.

[Response]: We have simplified our comparative analyses, and so only consider rates of change in a subset of cranial features versus all remaining parts of the skeleton. This sentence no longer appears in our revised text.

L330: ~~This quantifies the quantitative observation that major~~ Character transformations revealed by *Youngolepis* relate to the dentition and palate.

[Response]: This section has been revised in response to other suggestions, so this specific line no longer appears.

L333: ~~Should be~~ are.

[Response]: Revised.

L335: the sister lineage of ~~leading to~~ lungfishes.

[Response]: Revised.

L337: ~~The p~~Postcranial skeletal features of the earliest lungfishes are conservative.

[Response]: Revised.

L340: ~~arrangements~~ anatomy.

[Response]: Revised.

L346: ~~Porolepiformes~~ “Porolepiformes”.

[Response]: Revised.

L348: ~~more than~~ > 50 Ma.

[Response]: Revised.

L350: the first ~~stage~~ stages of which ~~is~~ are apparent in *Youngolepis*.

[Response]: Revised.

L687: Add ‘portion of the’.

[Response]: Added.

L705: Add ‘portion of the’.

[Response]: Added.

Reviewer #2

L.17: Durophagy becomes widespread only Devonian placoderms, especially in Middle and Late Devonian pachyosteomorph arthrodires, such as Mylostomatids. This is coeval with the rise of durophagous holocephalan chondrichthyans, which then will populate the Carboniferous marine ecosystems.

[Response]: Thank you for your comment. We discussed durophagy in Devonian placoderms in the Introduction.

L.53: *Youngolepis* also occurs in the Lower Devonian of northern Vietnam.

[Response]: Added.

L. 79: expansive...isn't it rather 'expanded'?

[Response]: Revised.

L. 94.: I have one question regarding the branchial apparatus: is there any evidence for branchial “teeth” or denticles? Such minute teeth covering the gill arches are present in porolepiforms, but lacking in lungfishes, and not shown in the scans of *Youngolepis* illustrated here. Such a lack of branchial teeth might be one more synapomorphy of the two taxa, as branchial teeth are widely present in all other osteichthyans.

[Response]: The present reconstructed 3D virtual models show that there is only a tooth plate on the basibranchial, but the other branchials do not bear teeth. We agree that the lack of branchial dentition is likely a character uniting *Youngolepis* and lungfishes, but we conservatively do not consider this trait in our analyses given that very small denticle patches might be near the limit of our scan resolution.

L. 219: Is it what Bjerring once called “hamuloquadrate”?

[Response]: Yes, it is.

L. 223: Palatal bite is clearly present in coelacanths (including extant ones), and was probably already there in porolepiforms, as suggested by the numerous toothed platelets covering the palate in *Glyptolepis* and *Powichthys* (see e.g. Clément & Janvier 2004, fig.8)

[Response]: Thank you for raising this important point. We have emphasized the significance of the early development of a palatal bite in dipnomorphs, as well as clearly discussing how the mechanism differs in porolepiforms and *Powichthys* on one hand (a field of many small toothplates) and *Youngolepis* and lungfishes on the other (expanded field of the entoptergoid bearing radiating tooth rows).

L.268: OK, and the same formation of Vietnam yealds toothplates that are almost identical to *Diabolepis*.

[Response]: We agree. These toothplates cannot be attributed to *Youngolepis*, but likely derive from *Diabolepis* or a *Diabolepis*-like taxon.

L.271: Interestingly, the prearticular of *Glyptolepis* shows an incipient radial patterning of the denticles, radiating from its posterior region (see Jarvik’s monograph).

[Response]: Thank you for sharing this interesting observation. We examined Jarvik (1972), but did not see the specimen showing this. The prearticulars most clearly illustrated (for *Holoptychius*: plate 32, fig. 1, plus restoration in fig. 47 of the main text; *Porolepis*: plate 12, fig. 2) do not seem to differ much in

the pattern of denticles from other sarcopterygians (i.e., they show a shagreen of denticles with no obvious pattern). Jarvik does not describe the prearticular in any particular detail in the text. Perhaps we missed the correct specimen; we are happy to mention this if we can track down the appropriate figure(s).

L. 695: it is rather 'medial' view.

[Response]: Revised.

Reviewer #3

This manuscript provides a thorough explanation about the morphology of *_Youngolepis_* species and, if I have understood correctly, how its specific and unique features seem to help better understand how the feeding mode evolved in bony fishes. While the authors have extensively documented the morphology of this species, the methods used to infer the morphological phylogeny and the divergence times are vaguely explained. There is a substantial amount of information that should be present both in the main and the supplementary text in case anyone was to reproduce the analyses mentioned in this manuscript.

[Response]: Thank you for the positive comments. We have taken steps to make our analyses more explicit and thus repeatable.

Below, you will find some suggestions with regards to the text as well as some questions I have about the methods used in this study. Hope that they are useful!

[Response]: Thank you for these suggestions.

===

Line 15: Maybe write "Innovations **about**" instead of "Innovations *~of~*".

[Response]: Revised.

Line 18-19: Maybe write as "New and exceptionally preserved *_Youngolepis_* specimens, which lived during the Early Devonian, [...]". I think it reads better for a non-familiar audience if written that way.

[Response]: Thanks for your revision. We rewrote the sentence as you suggested.

Line 25: Did you mean "pie*r*cing the prey"?

[Response]: Corrected.

Line 25: Rewrite to "total-evidence dating (TED) analysis". You might also want to add some citations here about previous TED analyses.

[Response]: We thank for the helpful suggestion. We modified the statement from "total-evidence dating analysis" to "Bayesian tip-dating analyses" because we didn't use molecular data in our analyses.

Line 26: When you say that you incorporate these new observations in a TED analysis, are the "observations" morphological data? If so, it might be good to clarify as either molecular and/or morphological data can be used to infer species divergence times in a TED analysis.

[Response]: Yes, we only used the morphological data in this work, and we clarified it in the reviewed version.

Line 49: "remain" in singular as it refers to "the timing and sequence".

[Response]: Yes, thank you for your correction. We revised it.

Line 49: When you say "a lack of information", what do you mean? Lack of evidence in the fossil record, incomplete fossils from different taxa that lead to missing characters in morphological matrices, ...? Please clarify here.

[Response]: We rewrote this sentence as 'However, the timing and sequence of the origins of these and other key feeding innovations remain unclear due to the incomplete fossils from the closest relatives of lungfishes that lead to missing characters in morphological matrices.'

Line 91: Just because I see you have been consistent with the oxford comma, you might want to add it here after "dermopalatines".

[Response]: Yes. We added the comma after 'dermopalatines'.

Line 196: Here, you are not only reporting the results you obtain when you infer a phylogeny (more specifically, a *morphological* phylogeny), but you are also reporting estimates of divergence times. Therefore, it would be fair to use a header such as "Morphological phylogeny inference and divergence-time estimation analysis". In addition, it would be good to mention the approach you have used to infer the morphological phylogeny and the divergence times so the reader can have an idea about what you have done before going to the "Methods" section.

[Response]: Following the referee's suggestions, we modified the header and added short notes of approaches we used for inferring the morphological phylogeny and the estimation of divergence times.

Line 380: You say "characters", but character data can be either morphological or molecular. It is best if you write "morphological character data" or "morphological data" instead of just "characters" when you first introduce the concept to the reader.

[Response]: We thank for the helpful suggestion, and we emphasized “morphological” characters in the method section.

Line 383: How many characters were removed and how many new characters were added based on your observations? You might want to add this information here. It would also be nice to include how many states the characters you include in the matrix have (e.g., maybe there are morphological characters with just 2 states and others with 5 states, so just write the range from the minimum to the maximum number of states one would expect to find in your morphological matrix).

[Response]: As the referee’s suggestion, we revised this part. We did not remove any characters, and we added 10 new characters based on our observations. The number of the character states range from 2 to 6.

Line 387: When using TNT, do you need to choose specific values for any parameters or do you need to choose specific options to run the software? If so, you might want to give more details about the setup that you have used to run this software to infer a morphological tree under parsimony so the user can reproduce your analysis.

[Response]: We agree with the referee and added the details of the setup in TNT in the revised manuscript as “A parsimony analysis was conducted in TNT 1.5. All of the characters were equally weighted. The maximum number of trees was set to 20,000 in memory. A traditional search with 1,000 replicates of Wagner trees using random additional sequences. The TBR branch swapping that held 20 trees per replicate was performed. All of the characters were equally weighted. Supports for the clade were evaluated by Bootstrap resampling using standard absolute frequencies (1,000 replicates) and Bremer decay indices retaining suboptimal trees up to 20 extra steps”.

Line 387: In addition, I would like to see if using a maximum-likelihood approach to infer the morphological phylogeny with your data (e.g., you can use either `RAxML-ng` or `IQ-TREE` for this analysis) converges with the estimated morphological tree inferred under parsimony. This is because the morphological matrix you have is not extremely large and an ML approach can run in a reasonable amount of time - besides, ML approaches have been widely used to infer both molecular and morphological phylogenies for the past years, more than parsimony has been used. For instance, if you choose to run `IQ-TREE`, you can follow the instructions [here] (<http://www.iqtree.org/doc/Tutorial>) under the section "Binary, morphological and SNP data", where you will find the commands you need to run when using your morphological matrix as the input data. You are given very clear guidelines about how you can choose the model under which you will infer your morphological phylogeny as well as the commands you need to use to account for acquisition bias.

[Response]: We agree with the referee that there are different methods for inferring phylogenies and the placement of *Youngolepis* should be tested with different methods. Following the referee's suggestions, we added maximum-likelihood phylogenetic analysis in our manuscript. We performed maximum-likelihood non-clock analysis in IQ-TREE (Supplementary Data 3). We used the Markov K variable model as the substitution model (with commands "MK+ASC"), and used a discrete gamma distribution with 4 categories to account the rate heterogeneity across characters (with command "G"). The maximum likelihood tree is provided in Supplementary Figure 3. Clade supports were examined with SH-aLRT branch test and Ultrafast Bootstrap with 1000 replicates for each. The results here are consistent with those of other analyses we present.

Line 391: Rewrite to "tip-dating analyses".

[Response]: Corrected.

Line 392: "phylogenetic dataset" is not the correct term you should use here. You are using two input files for each analysis (according to the supplementary material that I see you have attached): a "morphological character matrix" and a "parsimony-inferred morphological phylogeny". These are the input data that MrBayes is using to infer the divergence times (apart from the calibrations I see you have included and the different priors you have set) under a Bayesian tip-dating approach. Also, note that the format of the files in which these data are stored is nexus. Please, remove "phylogenetic dataset" and rewrite this bit accordingly.

[Response]: We have followed the referee's suggestion and changed "phylogenetic dataset" into "morphological data matrix".

Line 395: You write "The first did not apply any character partitioning, while the used four partitions". Do you mean that in "the first analysis" you did not partition your morphological character matrix while in "the second analysis" you partitioned it into four blocks based on anatomical systems? If this is what you meant, please rewrite this section so it is clearer for the reader. Also, it would be useful to know how many characters you included in each partition and the range of the states that each character had in each block (i.e., min-max range per block).

[Response]: We have followed the referee's suggestion and rewrote this section.

Specific comments about the "Methods" section:

- Why did you use `\prset clockratepr = normal (0.01, 0.02)`? I mean, why did you choose that specific distribution for the prior on the rates? You should include in the methods section a justification for choosing a normal prior distribution with this mean and sd in the same way that you have included the

details about the FBD prior you have used. The same applies for other priors you use when you run MrBayes.

[Response]: We agree with the referee's suggestions that this is an issue to be addressed. We reran all our tip-dating analyses with a more reasonable prior for clock rate in our refined manuscript. To derive the clock rate prior, we used R code that utilized the maximum likelihood non-clock tree as input, the estimated mean age for each taxon and an estimated root age. We first extracted the path length (morphological distance) from each tree tip to the tree root with the function "node.depth.edgelen" in package "ape". Next, each path length was scaled by the difference between estimated root age and the mean age for each tip. For the root age, we used a fixed age of 430 Ma, which is slightly older than the first appearance age of the outgroup taxon (Zhu et al., 2009). The first appearance and last appearance ages for each taxon were downloaded from THE DEEP BONE (Pan and Zhu, 2019), and the mean age for each taxon was estimated as the mean of these two values. Finally, the mean value of the scaled path length was used as the mean of the clock rate, and the exponential distribution (exponential (40)) with a similar mean was set as the prior of the clock rate. We used default setting (exponential (1)) as the prior of rate variance parameter.

Zhu, M. et al. The oldest articulated osteichthyan reveals mosaic gnathostome characters. *Nature*, 458(7237), 469-474 (2009).

Pan Z.-H., Zhu M. VPPDB: Hosting the Data for Vertebrate Paleoanthropology. *Acta Geologica Sinica (English Edition)*, 93 (supp.1): 61-63 (2019).

- Why did you choose to run an ILN instead of an autocorrelated-rates model? It would have been nice to see a model selection analysis with which the best rate model (ILN or the AR) is selected according to the data (and then you can use that when you infer the divergence times).

[Response]: We agree with the referee's suggestions; the patterns of morphological evolution and estimation of divergence time should be tested under different model settings. We added to our previous set of analyses. We first performed analyses of the unpartitioned matrix with different relaxed clock models including independent gamma rate (IGR), independent lognormal model (ILN), and TK02 (an autocorrelated-rates model, AR), and with the tree topology constrained as the parsimony strict consensus tree or unconstrained. Each analysis was run 200 million generation, and the first 30% samples were discarded as burn-in. Convergence of parameters was identified using Tracer (ESS > 200). The results are given in Supplementary Table 1. Analyses based on AR model did not converge (Supplementary Table 1), but all converged models exhibit similar tree shapes, divergence-time estimate, and inferred patterns in rates of character evolution over time. For partitioned analyses, we had initially performed analyses with four anatomical partitions (including external dermal bones of the skull, oral elements, neurocranium, and postcranium) matrix based on model settings mentioned above. However,

these analyses failed to converge. We applied a simplified, 2 partition approach (traits relevant to feeding, all remaining characters) and repeated all the analyses. Analyses based on AR model did not converge (Supplementary Table 1), and all converged models (ILN and IGR) yield similar tree shapes, divergence-time estimate, and inferred patterns in rates of character evolution over time. The divergence time estimates from partitioned and unpartitioned matrix were also comparable. Because all analyses under the AR models failed to converge, we did not perform any stepping-stone sampling to estimate marginal model likelihoods. Stepping-stone sampling typically takes 30 to 50 times longer using, say, 50 stones than a simple MCMC. All converged analyses support a similar picture of rates through time, evolutionary timescale, and the position and significance of *Youngolepis*, but we understand a desire to compare models. As an alternative, we examined the harmonic mean estimator from the MCMC as a rough guide. On this basis, we used the results of IGR model to make the text figure, but included all results in figures in the supplement.

- Are you using a specific fossil to set the exponential prior on the root age? If so, please specify which one so you can justify the age for the root.

[Response]: We used an offset exponential distribution as the prior of the root age with a minimum value of 427.95 Ma and a mean value of 430.46 Ma, which were slightly older than the outgroup taxa, *Guiyu oneiros*, in our matrix (Zhu et al., 2009).

Zhu, M. et al. The oldest articulated osteichthyan reveals mosaic gnathostome characters. *Nature*, 458(7237), 469-474 (2009).

- You have not confirmed if your MCMCs have converged. Common practice is to run each analysis at least twice (and long enough) to confirm that chains have converged. For instance, you can visually check if chains have converged by plotting the divergence times that you infer under the two runs and, if you see a line $x \approx y$, then the chains are likely to have converged. In addition, you should also verify that the effective sampling size (ESS) for each parameter inferred is $ESS \geq 200$. You can use R packages such as `coda` or `rstan` to verify this. You can also use the software `Tracer` for that purpose. If your chains have not converged, you might need to increase your sampling frequency while keeping the same number of samples to be collected (in that way, you do not increase the amount of disk space for the output file in which the collected samples are printed out but you are still running the chain longer). To calculate the sampling frequency and the length of the chain, you need to know that: $length_chain = burnin + sampling_freq * num_samples$

[Response]: We fully understand the referee's concern, and apologize for our lack of clarity in the previous version. We have rerun all our Bayesian tip-dating analyses with a more reasonable clock rate prior and added more sensitive analyses based on different relaxed clock models (IGR, ILN, and

AR). We have checked the convergence of parameters with Tracer (ESS > 200), and make sure all results reported in the main text were output from converged models (Supplementary Table 1). We also made this clear in our revised manuscript.

- In your supplementary material, you have four different directories: `tip dating_constrained` with strict consensus topology from parsimony analyses`, `tip dating_unconstraint`, `tip dating_4 partitions_constrained` with strict consensus topology from parsimony analyses`, and `tip dating_constrained` with majority consensus topology from parsimony analyses`. My questions and comments with regards to this zip file are the following:

a) Please rename the file names so there are no spaces (e.g., use underscores or dashes). It is then easier to deal with them if they are to be open from the terminal.

[Response]: We renamed the files (Supplementary Data 4).

b) Fix the typos in the file names, i.e., "tip dationg".

[Response]: Corrected.

c) Even though you have figures with the output morphological phylogeny drawn in your supplementary material, you should include a file with the resulting timetree inferred by MrBayes in the supplementary material either in nexus or newick format (these are the common formats used to read the trees in software such as `FigTree` or to manipulate them in `R` or `Python`; among many other uses). It would also be nice to see the output files by MrBayes and know which seed number was used to run the chain so the analyses can be reproduced by the user.

[Response]: We submitted majority rule consensus tree and files for testing parameter convergence output from Mrbayes (Supplementary Data 4).

d) I see in your nexus input files for MrBayes that you have used specific calibrations for the tip-dating analysis. Why did you specifically use uniform densities as calibrations and not, say, softbound calibrations as most dating analyses do? What are the maximum and minimum ages of each calibration you include based on? You should have a justification for each calibration included in the analysis (e.g., age justification based on a specific taxon?). You can add this information in the supplementary material.

[Response]: We apologise that this was not clear. The uniform distributions are used to specify uncertainty surrounding the age of terminal taxa, which are included in the analysis as OTUs rather than node constraints. The comment implies that our analyses used node calibrations (apart from the root node; see comments above). This is not correct.

The uniform distribution is appropriate here because we generally know a maximum and minimum possible age for a fossil (e.g., based on absolute age

estimates for the top and bottom of divisions of the geological record), but we generally do not have sufficient information to specify a distribution more informative than a uniform within those upper and lower (hard) bounds. The maximum and minimum age of each fossil taxon were taken from THE DEEP BONE. Application of uniform tip-age priors reflects standard practice for divergence-time estimation using fossil tips:

Fischer, V. *et al.* Extinction of fish-shaped marine reptiles associated with reduced evolutionary rates and global environmental volatility. *Nat Commun* 7, 10825 (2016).

<https://doi.org/10.1038/ncomms10825>

Brocklehurst, N., Benson, R.J. Multiple paths to morphological diversification during the origin of amniotes. *Nat Ecol Evol* 5, 1243–1249 (2021). <https://doi.org/10.1038/s41559-021-01516-x>

e) In your ms, you talk about divergence-time inference when using a topological constraint, yet here you have two directories with input data that suggest you ran two analyses without constraints. Why don't you report the results you obtained with the "unconstrained" analyses here in the main text? I can see you have Supp. Fig. 5, but you do not mention anything in the main text or compare the estimates you get when the topology was constrained and when it was not. Are there any topological rearrangements? Are times older or younger if constraints are/are not applied? Etc.

[Response]: We understand the referee's concerns. We supplemented the unconstrained analyses results (Supplementary Figs. 5,11). Please see them in the Supplementary material. The estimated time of the nodes stemward of *Diabolepis* is roughly similar between the unconstrained and constrained analyses. The estimated time of the nodes crownward of *Diabolepis* shows more variability because this clade has many unresolved polytomies. However, this does not affect the conclusions of this study because we are focused on rates on branches stemward of *Diabolepis*.

f) You provide an R script which should output some plots and CSV files, but I cannot see them in the supplementary information. Could you please attach them? Also, it would be nice if you provided the user with the input files you use to run this R script so they can reproduce your results.

[Response]: We provide the input files and output files in the Supplementary Data 4.

g) It would be interesting to plot the differences between the time estimates inferred when partitioning the data and when the data were not partitioned.

[Response]: We provide all the trees with divergence time of the divergence models (partitioned and unpartitioned) in the Supplementary material (Supplementary Figs. 4–17).

REVIEWER COMMENTS

Reviewer #1 (Remarks to the Author):

Thank you for your attention to the questions I raised in my initial review. You addressed all of them directly and satisfactorily. I look forward to your future work on these and related groups of sarcopterygians.

Reviewer #2 (Remarks to the Author):

The revised version of this manuscript is now excellent. The authors have correctly considered all the remarks made by the reviewers

Reviewer #3 (Remarks to the Author):

The new version of the manuscript has greatly improved. Nevertheless, I still have some suggestions and questions with regards to the analyses. Please see my comments below.

MAIN TEXT

=====

Line 403-404: Rewrite with the following and add if it was with or without replacement:
"The support values to assess branch supports were calculate by using a bootstrap approach resampling (with or without?) replacement.

Lines 409-414: Rewrite the whole section with the following:
"We also inferred the morphological tree under a maximum-likelihood approach using IQ-TREE (Supplementary Data 3). We used the M_k_model (Lewis, 2001) as the substitution model (Jukes-Cantor type model for morphological data, where M stands for "Markov" and _k_ for the number of possible states) and made sure to correct for acquisition bias (i.e., argument "MK+ASC" used). We also accounted for rate heterogeneity across characters by using the discrete Gamma model (Yang, 1994) with 4 rate categories (i.e., argument "G"). The best-scoring maximum-likelihood tree is provided in Supplementary Figure 3. We used the SH-like approximate likelihood ratio test (Guindon et al., 2010) and the ultrafast bootstrap with 1000 replicates to assess the support of the branching patterns estimated in the phylogeny. The results here are consistent with those of other analyses we present."

References that you need to include:

Paul O. Lewis, A Likelihood Approach to Estimating Phylogeny from Discrete Morphological Character Data, *Systematic Biology*, Volume 50, Issue 6, 1 November 2001, Pages 913–925, <https://doi.org/10.1080/106351501753462876>

Yang, Z. Maximum likelihood phylogenetic estimation from DNA sequences with variable rates over sites: Approximate methods. *J Mol Evol* 39, 306–314 (1994). <https://doi.org/10.1007/BF00160154>
<https://link.springer.com/article/10.1007/BF00160154>

Stéphane Guindon, Jean-François Dufayard, Vincent Lefort, Maria Anisimova, Wim Hordijk, Olivier Gascuel, New Algorithms and Methods to Estimate Maximum-Likelihood Phylogenies: Assessing the Performance of PhyML 3.0, *Systematic Biology*, Volume 59, Issue 3, May 2010, Pages 307–321, <https://doi.org/10.1093/sysbio/syq010>

Line 417: Space in "datamatrix", rewrite as "data matrix".

Line 421: Rewrite as: "We used the Markov variable model (Mkv)".

Line 424: Rewrite as: "allow *for* rate variation across".

Line 425-426: Rewrite as: "we used an in-house R script".

Line 426-427: Rewrite as: "the best-scoring maximum-likelihood morphological tree as input".

Line 428: After "and an estimated root age", add ", 430 Ma, based on the first appearance age of the outgroup taxon". I have moved the sentence in lines 431 and 432 here.

Line 428-429: Rewrite as: "We estimate the morphological distance from the tips to the root of the tree, the "path length", using the function `node.depth.edgelen`, part of the `ape` R package".

Line 430: Add "the": "*the* estimated root".

Lines 431-432: Delete from "For the root age" until "outgroup taxon". This sentence is included in my comment above.

Lines 432-434: In your letter, you wrote:

"The first appearance and last appearance ages for each taxon were download from THE DEEP BONE (Pan and Zhu, 2019), and the mean age for each taxon was estimated as the mean of these two values."

In the new version of your ms, you have:

"The mean age for each taxon was estimated as the mean of the first appearance age and the last appearance age".

I guess that you want to keep the version in the letter, so you might want to update the ms.

Line 438: Add the following: "*set* as the prior of the clock rate. We used *the* default settings [...]".

Line 442-443: Rewrite as "We used a uniform distribution, $U(0,10)$, as the prior of net speciation rate and a beta distribution, $Beta(1,1)$ [...]".

Lines 449-450: You say that the root age had "an offset exponential prior with a mean of 430.46 Ma" but then you say it had a "minimum of 427.95 Ma". This is confusing as it seems you were using a uniform distribution with max. 430.46 and min. 427.95. Could you clarify the text?

Line 459: Did you run at least two chains for each partition? In the way that you have written this, it seems you run each partition only once, and hence it is not possible to check for chain convergence. You might have assessed the ESS, but you still have not assessed chain convergence if you have not run each analysis at least twice. By looking at your supp. data, I can see you have files "xxxrun1.p" and "xxxrun2.p". Am I correct in thinking that you have run the analyses for each dataset twice? If so, please make sure to clarify in the main text and in the supplementary material section as it is not clear.

It is sometimes difficult for analyses running under the AR to converge because they need to run longer than IR models -- it is more difficult to detect autocorrelation!

If you ran the chains longer (i.e., keep the same number of samples, I suggest around 20,000 if 10,000 was not enough, but increase the sampling frequency) you will likely see the chains convergence. Make sure you run each analysis at least twice to assess chain convergence. You can also evaluate chain convergence by plotting the estimated mean times for each node in one run against those estimated in the second run.

Once the chains converge, you should be able to run a Bayesian model selection analysis.

I do not support the usage of the harmonic mean estimator as this is a very bad estimator of the marginal likelihood: it basically overestimates the marginal likelihood (see further comments in Xie et al. 2011, <https://academic.oup.com/sysbio/article/60/2/150/2461669>). You can remove this from Supp. Table S1.

SUPP MATERIAL

=====

Line 52: Rewrite as: "We first performed Bayesian *tip-dating* analyses".

Lines 73-75: Remove the text. The HME (harmonic mean estimator) should not be used as it tends to overestimate the marginal likelihood (see comments and paper mentioned above).

REVIEWER COMMENTS

Reviewer #1 (Remarks to the Author):

Thank you for your attention to the questions I raised in my initial review. You addressed all of them directly and satisfactorily. I look forward to your future work on these and related groups of sarcopterygians.

[Response]: Thank you for your evaluation, and for your helpful comments on a previous version that strengthened this contribution.

Reviewer #2 (Remarks to the Author):

The revised version of this manuscript is now excellent. The authors have correctly considered all the remarks made by the reviewers.

[Response]: Thank you for your evaluation, and for your helpful comments on a previous version that strengthened this contribution.

Reviewer #3 (Remarks to the Author):

The new version of the manuscript has greatly improved. Nevertheless, I still have some suggestions and questions with regards to the analyses. Please see my comments below.

[Response]: Thank you for these comments, which help to clarify aspects of our analyses. Please see our responses below.

MAIN TEXT

=====

Line 403-404: Rewrite with the following and add if it was with or without replacement: The support values to assess branch supports were calculate by using a bootstrap approach resampling (with or without?) replacement.

[Response]: We now explicitly note that this bootstrapping procedure involves sampling with replacement.

Lines 409-414: Rewrite the whole section with the following:

“We also inferred the morphological tree under a maximum-likelihood approach using IQ-TREE (Supplementary Data 3). We used the M_k_ model (Lewis, 2001) as the substitution model (Jukes-Cantor type model for morphological data, where M stands for "Markov" and _k_ for the number of possible states) and made sure to correct for acquisition bias (i.e., argument "MK+ASC" used). We also accounted for rate heterogeneity across characters by using the discrete Gamma model (Yang, 1994) with 4 rate categories (i.e., argument "G"). The best-scoring maximum-likelihood tree is provided in Supplementary Figure 3. We used the SH-like approximate likelihood ratio test (Guindon et al., 2010) and the ultrafast bootstrap with 1000 replicates to assess the support of the branching patterns estimated in the phylogeny. The results here are consistent with those of other analyses we present.”
References that you need to include:

Paul O. Lewis, A Likelihood Approach to Estimating Phylogeny from Discrete Morphological Character Data, Systematic Biology, Volume 50, Issue 6, 1 November 2001, Pages 913–925, <https://doi.org/10.1080/106351501753462876>

Yang, Z. Maximum likelihood phylogenetic estimation from DNA sequences with variable rates over sites: Approximate methods. J Mol Evol 39, 306–314 (1994). <https://doi.org/10.1007/BF00160154><https://link.springer.com/article/10.1007/BF00160154>

Stéphane Guindon, Jean-François Dufayard, Vincent Lefort, Maria Anisimova, Wim Hordijk, Olivier Gascuel, New Algorithms and Methods to Estimate Maximum-Likelihood Phylogenies: Assessing the Performance of PhyML 3.0, Systematic Biology, Volume 59, Issue 3, May 2010, Pages 307–321, <https://doi.org/10.1093/sysbio/syq010>

[Response]: Thank you for your revision. We revised the manuscript as you suggested.

Line 417: Space in "datamatrix", rewrite as "data matrix".

[Response]: Revised.

Line 421: Rewrite as: "We used the Markov variable model (Mkv)".

[Response]: Rewritten following suggestion.

Line 424: Rewrite as: "allow *for* rate variation across".

[Response]: Revised.

Line 425-426: Rewrite as: "we used an in-house R script".

[Response]: Revised.

Line 426-427: Rewrite as: "the best-scoring maximum-likelihood morphological tree as input".

[Response]: Rewritten.

Line 428: After "and an estimated root age", add ", 430 Ma, based on the first appearance age of the outgroup taxon". I have moved the sentence in lines 431 and 432 here.

[Response]: Revised.

Line 428-429: Rewrite as: "We estimate the morphological distance from the tips to the root of the tree, the "path length", using the function `node.depth.edgelen`, part of the `ape` R package".

[Response]: Rewritten.

Line 430: Add "the": "**the* estimated root".

[Response]: Added.

Lines 431-432: Delete from "For the root age" until "outgroup taxon". This sentence is included in my comment above.

[Response]: Deleted.

Lines 432-434: In your letter, you wrote:

"The first appearance and last appearance ages for each taxon were download from THE DEEP BONE (Pan and Zhu, 2019), and the mean age for each taxon was estimated as the mean of these two values."

In the new version of your ms, you have:

"The mean age for each taxon was estimated as the mean of the first appearance age and the last appearance age".

I guess that you want to keep the version in the letter, so you might want to update the ms.

[Response]: Yes, you are right. We updated it in the manuscript.

Line 438: Add the following: "*set* as the prior of the clock rate. We used *the* default settings [...]".

[Response]: Added.

Line 442-443: Rewrite as "We used a uniform distribution, $U(0,10)$, as the prior of net speciation rate and a beta distribution, $Beta(1,1)$ [...]".

[Response]: Rewritten.

Lines 449-450: You say that the root age had "an offset exponential prior with a mean of 430.46 Ma" but then you say it had a "minimum of 427.95 Ma". This is confusing as it seems you were using a uniform distribution with max. 430.46 and min. 427.95. Could you clarify the text?

[Response]: The root age had an offset exponential prior with a minimum of 427.95 Ma, which was slightly older than the time range of the outgroup taxon (423.0 Ma), and a mean of 432.95 Ma that was slightly older than the minimum. We did not set a maximum because there is no reliable reference. We did not use the uniform distribution, which would require us to make an *ad hoc* claim about a firm upper age limit. Meanwhile, we found that the previous mean root age (430.46 Ma) was incorrect and revised it accordingly.

Line 459: Did you run at least two chains for each partition? In the way that you have written this, it seems you run each partition only once, and hence it is not possible to check for chain convergence. You might have assessed the ESS, but you still have not assessed chain convergence if you have not run each analysis at least twice. By looking at your supp. data, I can see you have files "xxxrun1.p" and "xxxxrun2.p". Am I correct in thinking that you have run the analyses for each dataset twice?

If so, please make sure to clarify in the main text and in the supplementary material section as it is not clear.

[Response]: Yes, you are correct. We have run the analyses for each dataset twice. We added the clarification in the main text (line 424, version without markup) and in the supplementary material (line 56, version without markup). We performed Bayesian tip-dating analyses in Mrbayes 3.2.8. and each analysis computed two runs with four chains (one cold chain and three incrementally heated chains). The “xxxrun1.p” and “xxxrun2.p” files are the results of cold chains. We have checked the convergence of parameters with Tracer (ESS > 200), and make sure all results reported in the main text were output from converged models (Supplementary Table 1).

Line 459: It is sometimes difficult for analyses running under the AR to converge because they need to run longer than IR models -- it is more difficult to detect autocorrelation! If you ran the chains longer (i.e., keep the same number of samples, I suggest around 20,000 if 10,000 was not enough, but increase the sampling frequency) you will likely see the chains convergence. Make sure you run each analysis at least twice to assess chain convergence. You can also evaluate chain convergence by plotting the estimated mean times for each node in one run against those estimated in the second run. Once the chains converge, you should be able to run a Bayesian model selection analysis.

I do not support the usage of the harmonic mean estimator as this is a very bad estimator of the marginal likelihood: it basically overestimates the marginal likelihood (see further comments in Xie et al. 2011, <https://academic.oup.com/sysbio/article/60/2/150/2461669>). You can remove this from Supp. Table S1.

[Response]: As the referee suggested, we reran the analyses under AR model with doubled generations (400 million) and sampling frequency (20,000). Although each analysis spent a week, they did not converge (Supplementary Data_4). ESS was very poor (Supplementary Data_4). Generally, the phylogenetic relationships of lungfishes are not stable, especially among some Early Devonian forms. Thus, this outcome seems reasonable. The current conditions do not allow us to run longer chains of AR because each analysis takes two weeks or more and does not seem to yield convergent results in a limited time. Even if we obtain convergent results, subsequent Bayesian model selection analyses will take at least half a year. Moreover, both converged analyses (IGR and ILN) support a similar picture of evolutionary rates, timescales, and the position of *Youngolepis*. Hence, the particular model favored by a model selection procedure does not substantially affect the conclusions of this study and its implication for the evolution of key anatomical features of lungfishes demonstrated by our new fossils of *Youngolepis*. Certainly, we understand a desire to compare models, and we are very willing to do this in future studies. As you suggested, we deleted the harmonic mean estimator in the Supp. Table S1.

SUPP MATERIAL

=====

Line 52: Rewrite as: "We first performed Bayesian *tip-dating* analyses".

[Response]: Revised.

Lines 73-75: Remove the text. The HME (harmonic mean estimator) should not be used as it tends to overestimate the marginal likelihood (see comments and paper mentioned above).

[Response]: Deleted.